# Global extent and drivers of mammal population declines in protected areas under illegal hunting pressure

**Alfan A. Rija**[1]*, **Rob Critchlow**[2], **Chris D. Thomas**[2], **Colin M. Beale**[2]

**1** Department of Wildlife Management, Sokoine University of Agriculture, Morogoro, Tanzania, **2** Department of Biology, University of York, York, United Kingdom

* al.rija10@gmail.com

**Data Availability Statement:** Data are available as Supporting Information files.

**Funding:** AAR was a commonwealth scholar funded by the U.K. government. The sponsor-Commonwealth Scholarship Commission had no

## Abstract

Illegal hunting is a persistent problem in many protected areas, but an overview of the extent of this problem and its impact on wildlife is lacking. We reviewed 40 years (1980–2020) of global research to examine the spatial distribution of research and socio-ecological factors influencing population decline within protected areas under illegal hunting pressure. From 81 papers reporting 988 species/site combinations, 294 mammal species were reported to have been illegally hunted from 155 protected areas across 48 countries. Research in illegal hunting has increased substantially during the review period and showed biases towards strictly protected areas and the African continent. Population declines were most frequent in countries with a low human development index, particularly in strict protected areas and for species with a body mass over 100 kg. Our results provide evidence that illegal hunting is most likely to cause declines of large-bodied species in protected areas of resource-poor countries regardless of protected area conservation status. Given the growing pressures of illegal hunting, increased investments in people's development and additional conservation efforts such as improving anti-poaching strategies and conservation resources in terms of improving funding and personnel directed at this problem are a growing priority.

## Introduction

Protected areas (PAs) are the cornerstone of biodiversity conservation and have increased in number globally (covering at least 15% of global terrestrial land and 7% of ocean [1, 2]; to reach estimated conservation targets of protecting at least 17% of the global land area by 2020 [3, 4]. The effectiveness of PAs is challenged by increasing pressures from management efficiency such as governance and resource deficiency [5–7], and anthropogenic pressures such as habitat loss and illegal hunting [8–10]. Consequently, many PAs continue to experience wildlife declines [2, 9], suggesting that enlarging the PA network alone does not necessarily lead to conservation success when other efforts such as improved law enforcement, funding and regulation of land use change pressure outside are not considered [2]. Despite these challenges to protecting biodiversity, broad scale patterns on illegal hunting in PAs and the consequences

role in the design, data collection and analysis, decision to publish or preparation of the manuscript.

**Competing interests:** The authors have declared that no competing interests exist

for species conservation remain poorly assessed, yet could help improve the effectiveness of PAs and improve biodiversity conservation [8].

Anthropogenic threats to wildlife within and outside PAs have been studied extensively including logging [11, 12], hunting [9, 13] and encroachment through cropland conversion and livestock grazing [8, 11]. These threats can have serious conservation and ecological implications. For example, recent assessments of wildlife abundance across tropical forests reported at least 40% decline in mammal distribution range due to hunting [14]. Mammal population declines by 80% and 30% have been reported across tropical forests caused by hunting and land use change respectively [9, 15], consequently leading to potential loss of species interaction networks [13, 16]. Further, bushmeat hunting (i.e. hunting for wild meat) is a growing conservation problem across the savanna biomes [17], threatening several hundred thousand vertebrate species across the globe [13, 18]. Very few of those studies that investigate hunting impact account for protection status of an area and separate legal and illegal hunting, which may occur together [8, 9]. This potentially confounds efforts needed to target illegal hunting pressures particularly in PAs managed through law enforcement. At the individual PA level, the drivers of illegal hunting are often known. This includes subsistence hunting to ensure food security and offtake for commercial gains [17, 19]. Species traits such as taste [13, 20] and body size also influence hunting preference [9, 13, 20], but have rarely been assessed on a large scale [15]. Increasing access to PAs due to anthropogenic encroachment and land use change has also resulted in wildlife population declines, threatening PA effectiveness [2, 9]. Thus, research targeting the effectiveness of species conservation in relation to illegal hunting pressure is needed as PAs remain the major stronghold of biodiversity, and as important key refuge areas [8, 21].

Mammals contribute the highest proportion to total biomass across forest and savanna landscapes making them highly sought after by illegal hunters [19, 22, 23]. Loss of mammals due to illegal hunting pressure has been related to substantial loss of important functional characteristics such as seed dispersal and regeneration and ecological interactions, thus endangering many ecological services that they support [2, 24, 25] and threatening humanity [26]. Therefore, due to their functional importance and growing threats, mammals are an excellent taxon to document hunting pressures and identify improved conservation strategies in PAs. Fortunately, the distribution changes of mammals brought about by hunting are broadly known [14, 27]–but current research rarely focuses on PAs [8].

Few studies have quantified the relative contribution of individual threats to the overall pattern of population change and decline in PAs [10]. Such an assessment is required to identify strategies to improve PA effectiveness, such as where to target additional resources and which actions are most effective at enforcing existing regulations [28]. Here, we investigated spatial patterns and impact of illegal hunting on wild mammal populations within PAs using a comprehensive database collated from 40 years of published literature. We built models that integrate both the outcome of illegal hunting pressure on species and the socio-ecological parameters to derive patterns of impacts on broader regional and global scales. Specifically, we aimed to:

I.  Assess the extent and scale of research on illegal hunting of mammal populations in PAs.

II.  Examine what factors are associated with mammal declines and their variations across species and PAs and between geographic regions.

III.  Identify current limitations in existing literature and propose recommendations for improving PAs effectiveness in relation to illegal hunting of mammals.

## Material and methods

### Data collection

Literature searches were carried out in Web of Science, Google Scholar and Scopus between March 2014 and 2015 and in March 2020. Restricting the search period to 1980–2020, the search terms included: 'illegal activity' OR 'illegal activities' OR 'illegal hunting' OR poach* AND 'protected area' OR reserve OR 'biodiversity outcome'. Full details of search terms can be found in the PART II in S1 File. Results were screened based on criteria (i-iv below) to ensure each paper were related to both PAs and illegal mammal hunting:

i. Whether the research was done in a PA and addressed issues of illegal hunting of mammals.

ii. If the research showed impact on species population within the PA

iii. Only used primary data papers not meta-analyses or reviews.

iv. For results that returned more than one study covering the same PA and year of data collection these were examined for relevance and only one that satisfied all criteria (i-iii) was included in the analysis.

From each paper, we extracted information on PA location, threat types, study species, reason for illegal hunting, purpose of research and data collection method (Table 1 and S1 Raw data). We recorded population trend (i.e. decline, no decline or unstated) for each PA/species combination and the reasons mentioned for such outcomes if not directly related to illegal hunting. Each species / site combination and associated variables became one row in the dataset, including the impact scores (1 = species decline or 0 = no decline or NA = no reported impact for that species) and any covariate information (see S1 Raw data for detailed database). The method arriving at the reported population trend status was recorded but not analyzed because most studies did not show the data used to arrive at a species outcome (e.g. decline), thus was difficult to tease apart whether the species decline was causal or correlative, a common problem in many meta-analysis studies [29].

Species body size is likely to influence mammal hunting risk [13]. We extracted body mass from the mammal database PanTHERIA (www.pantheria.org) and EltonTraits [30]. The body mass for two species (*Eulemur rufifrons* and *Gazella dorcas massaesyla*) were extracted from published literature (PART I in S1 File). The WCMC IUCN Protected Planet database was used to identify the geographic location (latitude/longitude) of a PA. To assess whether legal

**Table 1. Description of the terms under methods extracted from the reviewed papers reported in Fig 2.**

| Method | Description |
|---|---|
| Animal counts | Population size and trend of animals in a protected area under illegal hunting pressure documented from animal count methods such as aerial sample counts, systematic strip transect survey and Distance sampling. |
| Interviews/ counts | Illegal hunting and impact on mammal population trends from interviews of field rangers or local communities and direct animal counts in the field. |
| Interviews | Assessment of illegal hunting and trend of illegally hunted populations from local communities directly involved in illegal hunting or consumption of bushmeat. E.g. use of questionnaires and interviews surveys. |
| Market Surveys | Records of live animals or carcasses from markets, or surveys of markets to record these data |
| Other surveys | Surveys of illegally killed mammal carcasses in the field, surveys of snares used in illegal hunting, animal bone surveys, poacher arrest records, combined carcass survey and local expert opinion of the population trends of illegal hunted animals. |
| Patrol data | Information about illegal hunting of wildlife and impact on population trends inferred from law enforcement data conducted by rangers in a protected area. |

status influences the outcome of species populations, we extracted the IUCN category of each PA [31].To explore the effect of legal status on species decline we grouped the PAs into two levels of protection: strict PAs (IUCN category I & II) and less strict PAs (categories III-IV). PAs in IUCN categories V-VI were excluded from analysis because they are not specifically designated for biodiversity conservation [32].

To assess whether population change reported was related to broad scale economic or social change, we used country-level human development index (HDI) and agricultural land use change (ALC) index extracted from the UNDP and World Bank databases [33, 34]. Agricultural land use change index is a measure of the amount of land converted to agriculture and other human activities such as settlement. The ALC index is associated with habitat loss within PAs due to encroaching on PAs [2, 11, 35], and is also related to illegal hunting because farmers encroaching on PAs often engage in illegal hunting of animals [36]. We calculated the ALC over a decade period encompassing the times when research for the reviewed papers were conducted as most papers did not report the exact dates of data collection. We used HDI as a measure of socio-economic change and governmental effectiveness because HDI is a direct indicator of development [37].

## Data analysis

We used each unique combination of species within an individual PA (i.e. species × PA) reported in a paper as a study and as an individual data point, with therefore potentially several studies per paper (see data extraction in methods and S1 Raw data). Using PA location and publication year, we assessed spatial and temporal patterns of illegal hunting in PAs. Prior to conducting formal analyses, species body mass (BM) was logged and scaled and ALC scaled. HDI was used than other variables such as country corruption level and, government effectiveness because HDI has mostly been used in similar assessments and is a direct measure of poverty [38, 39]. To examine the effect of species traits and socio-economic variables on illegal mammal hunting in PAs, we used generalized linear mixed models (GLMMs) with a binomial error term and logit link function implemented in the R-package 'lme4' [40] in the statistical software R (version 3.6.3). For all models, we excluded records where the population outcome was unknown, and the response variable was whether a species had declined in that PA (1) or not (0). We built an initial global model incorporating six fixed factors: HDI, ALC, log species body mass, illegal hunting type (commercial, subsistence, or combined), PA protection status (i.e. IUCN categories classified as strict and less strict) and continent (Africa/Asia/Central and South America). Records from Europe were excluded from all analyses due to a low number of records (n = 3). Because different species could relate to the same PA and country as studies from other papers at different times, we accounted for this by including country, paper and PA as random effects in all models. We used a backwards stepwise removal of non-significant terms (with Chi-test) to evaluate the relative effect of each factor on the population decline. We obtained model confidence intervals around variables showing statistical significance in the minimum adequate model using the Wald-method [40]. Models of all analysis subsets were assessed using Akaike information criterion (AIC) to select the most parsimonious model after the stepwise removal of terms. We calculated a measure of variance explained by the models using the squared correlation between the response variable and the predicted values. These pseudo R2 values should be interpreted with caution as it only shows the variance explained by the fixed effects in the final model. Conditional R2 values were calculated using the MuMIn R package [41].

The majority of illegal mammal hunting records were from Africa and Asia. We therefore built two additional models using the same structure as the global model, but restricted data to

Africa and Asia respectively. Finally, because the savanna elephant (*Loxodonta africana*), had a large percentage of records (19%) in the Africa dataset, reflecting the increasing concerns for illegal hunting of elephant and illegal ivory trade [42], we built two further models for Africa using the same random effects structure as the global model: one excluding savanna elephants (where the fixed effects were HDI, ALC, log body mass and PA protection status) and another only including savanna elephants (where the fixed effects were HDI, ALC and PA protection status.

## Results

### Extent and scale of research on illegal hunting of mammals in PAs

Our searches found 2245 papers in total, 81 of which qualified our selection criteria for inclusion in the review (PART III in S1 File). The reviewed papers were from 48 countries and four continents, Africa, South America, Asia and Europe and covered 155 PAs (Fig 1). Further, the reviewed papers reported 294 mammal species to have been illegally extracted from the PAs across the four continents (PART IV in S1 File).

There was an increasing trend in research on illegal hunting in PAs with the number of publications increasing two-fold each decade since 1980. Types of study methods have also increased, particularly studies that have included use of ranger patrol data and interview techniques (Fig 2).

Most papers focused on single PA (i.e. local scale, n = 44), compared to PAs existing as one contiguous ecosystem (n = 24) or landscape (n = 19). All protected area types were investigated but the IUCN category II level of protection was researched the most (55.2%, n = 90). The research had varying purposes: investigating impacts on species (n = 64); conservation rationale (e.g. providing new methods for investigating illegal activities; n = 17) and management of

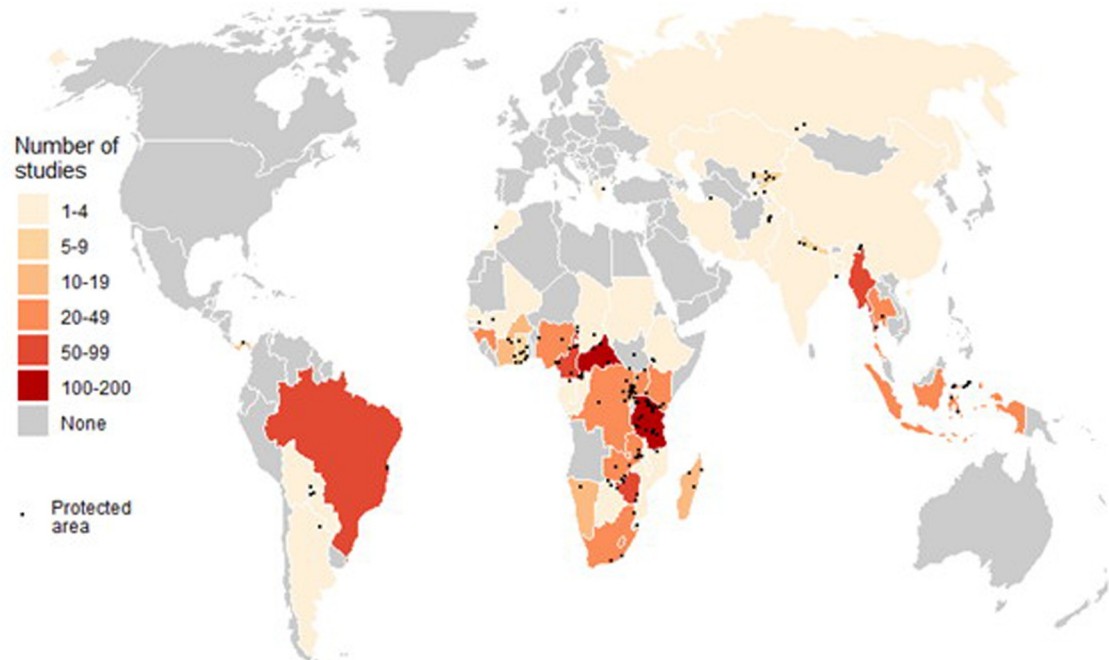

**Fig 1. Spatial distribution of research on illegal hunting of mammals in 155 PAs from 48 countries over four decades as collated in the literature.** Black dots correspond to the centroid of a PA where research for the reviewed papers was conducted. [The map used in this figure was sourced from Natural Earth, which is an open access map source].

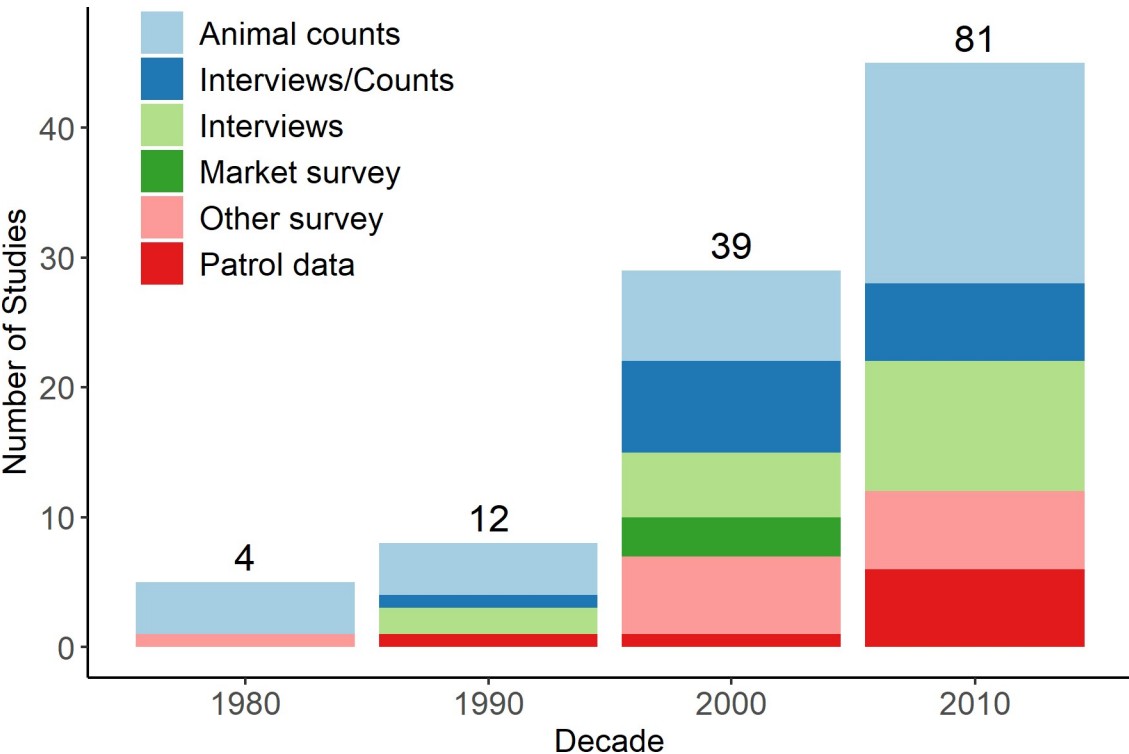

**Fig 2. Number of studies per data collection method over 10-year periods.** Numbers at top of bars indicate the cumulative number of publications. The majority (81%) of PAs studied were in the strictest IUCN categories (I-II).

illegal activities (n = 4). The publication trend has increased substantially during the last 40 years with a greater number of published papers since 2005; most of this increase was in Africa.

**Socio-economic and species traits influencing population decline in PAs.** We found differing impacts of socio-economic variables and species traits on the population trends of mammals in PAs as reported in final models that only included significant terms. In the global model (n = 523), species body mass and PA strictness had a significant effect on the probability of species decline in PA. The probability of a population to decline in a PA increased with body mass and populations in stricter PAs (IUCN categories I-II) had a lower probability of decline compared to less strict PAs (IUCN categories III-IV) (Fig 3 and Table 2; model1). When limiting analysis to Africa (n = 374), populations were more likely to decline in countries with a lower HDI and for species with greater body mass (Fig 4 and Table 2; model 2). Accounting for the influence of studies on illegal elephant hunting, results limited to Africa and excluding elephants (n = 302) showed that mammals with larger body mass were at greater risk of population decline in PAs (Table 2; model 3). Restricting analysis to African elephants (n = 72), showed that probability of population decline decreased with an increase in HDI (Table 2; model 4).

Results focusing on illegal mammals hunting in Asia (n = 65) showed that mammal populations in strictest PAs designated to protect biodiversity had a significantly greater probability of decline compared to mammal populations in less strict PAs (Fig 5 and Table 2; model 5).

## Discussion

We analyzed data published since 1980 to understand impacts of illegal hunting on species population decline in PAs. There was strong geographic bias in research on illegal hunting

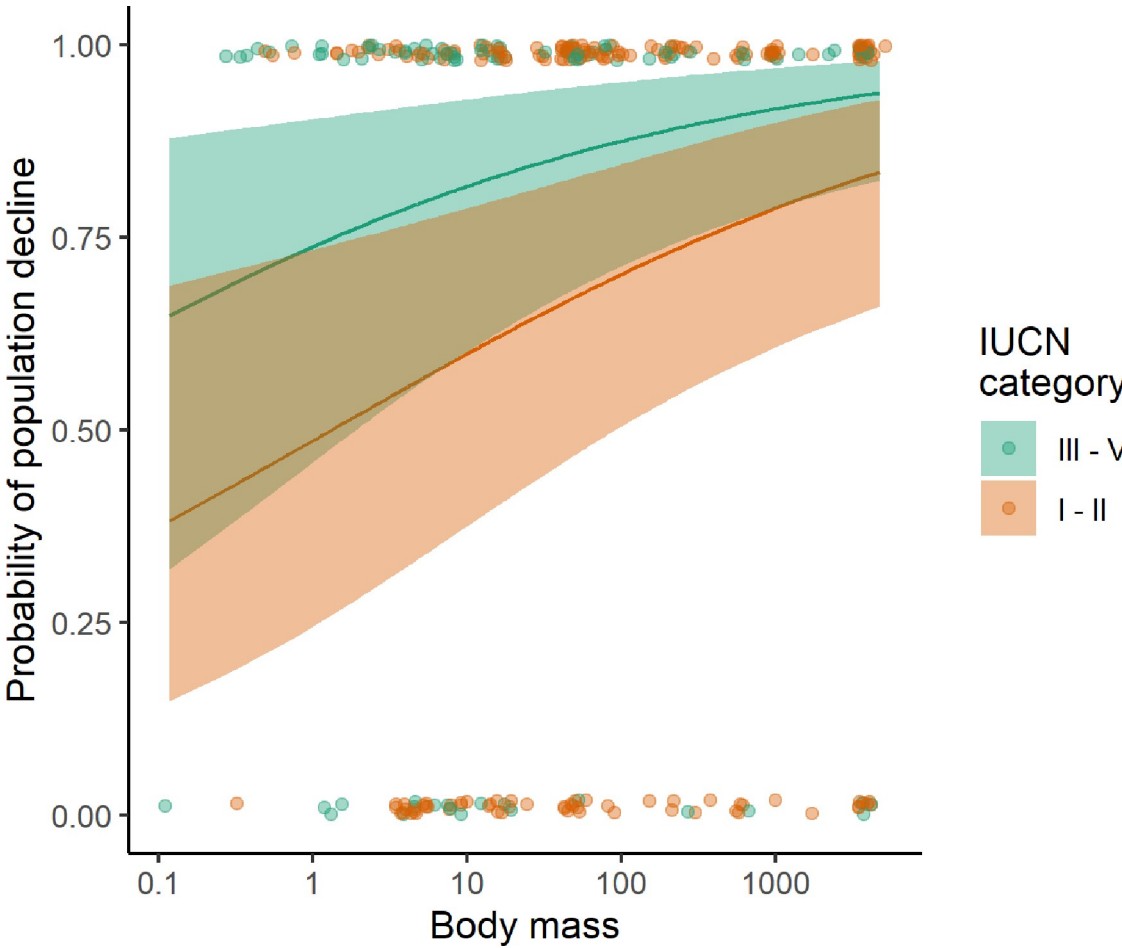

**Fig 3. Effect of species body mass and PA strictness on the probability of decline of mammal species from global records.**
Shaded area shows 95% confidence intervals. Circles represent raw data points.

within PAs dominated by records in Africa and Asia. Globally, we found that illegally hunted mammals in strict PAs (IUCN categories I-II) had a lower probability of population decline compared to less strict PAs (IUCN categories III-IV). In contrast, we found illegally hunted Asian mammals in strict PAs had a greater probability of population decline than less strict PAs. Human development index had a strong influence on likelihood of mammal declines in Africa, while larger bodied mammals were more likely to show population decline globally and in Africa.

The identification of correlations between human development indices and illegal hunting in this study supports a widely held view, e.g. [43–45] but one that is often based on limited data [46]: that biodiversity decline is higher in relatively poor regions. Low HDI scores could impact illegal hunting in two ways: firstly, poor people may tend to exploit species illegally from PAs because they have limited alternatives [47]. Secondly, poor countries have fewer resources to invest in PA conservation, therefore underfunding may result in increased illegal hunting, as well as other illegal activities such as agricultural encroachment in PAs due to insufficient law enforcement [48]. Hilborn, and others [49] demonstrated that increased funding budgets for anti-poaching activities in the Serengeti National Park greatly reduced illegal hunting pressures and led to the recovery of the buffalo population. However, increasing

**Table 2. Summary of GLMMs fixed effects of socio-economic factors and species traits on the probability of mammal decline in PAs.**

| Models | 1 | 2 | 3 | 4 | 5 |
|---|---|---|---|---|---|
| | Global | Africa only | Africa- ex.elephant | Africa- elephant only | Asia only |
| HDI | | -8.97* | | -15.80** | |
| | | (-16.59, -1.35) | | (-26.33, -5.28) | |
| Body mass | 0.21** | 0.34*** | 0.37*** | | |
| | (0.07, 0.34) | (0.16, 0.53) | (0.17, 0.58) | | |
| PA strictness | -1.11* | | | | 1.77** |
| | (-1.95, -0.27) | | | | (0.66, 2.88) |
| Intercept | 1.00 | 3.53 | -0.08 | 8.28*** | -0.35 |
| | (-0.22, 2.21) | (-0.11, 7.17) | (-1.32, 1.16) | (3.36, 13.14) | (-1.09, 0.39) |
| Observations | 523 | 374 | 302 | 72 | 65 |
| Pseudo $R^2$ | 0.37 | 0.47 | 0.43 | 0.68 | 0.16 |
| Conditional $R^2$ | 0.64 | 0.77 | 0.61 | 0.83 | 0.19 |

Significance *p<0.05;

**p<0.01;

***p<0.001.

Values in brackets represent 95% confidence intervals.

conservation funding may not necessarily result in improved conservation particularly when social and political constraints exist. For example, social and political unrest may increase rates of illegal hunting and encroachment in PAs, reduce wildlife populations and thwart conservation efforts altogether [50, 51]. Our results provide evidence that poverty, in as much as it is measured by the HDI, may have significant negative impacts on species due to accelerated illegal hunting, whether that be because of increased external pressures on PAs or decreased policing and protection within strict PAs.

Our model that included only African elephants also highlights the importance of low HDI in relation to illegal ivory trade. Poor people in countries with low HDI may directly engage in the ivory trade chain mainly as illegal hunters, supplying ivory to the demand illegal markets [52]. Such a trade chain may be maintained by ivory traders or consumers through providing means and or financing illegal hunters to target the protected areas [53, 54]. Additionally, people faced with poverty in low HDI countries may be forced to alternatives such as illegal hunting and habitat destructive activities [55]. Although we did not find an association of habitat loss and mammal declines, destructive activities such as charcoal burning and logging for timber and fuelwood is common in poor countries with low HDI [56, 57],thereby reducing habitat suitability for elephants and other mammals, leading to population declines [39, 58]. Our study highlights the need to consider human development issues more seriously to ensure effective conservation of biodiversity within existing PAs.

Large bodied species are likely highly susceptible to decline because they have slow growth rates and so overharvesting is likely to cause population decline [15, 59]. This is because low population growth rates in combination with illegal hunting are known to cause significant reduction in population persistence [60, 61]. An alternative explanation to the large mammal decline observed in our data could be that large mammals, due to their relatively bulk meat content compared to smaller mammals they are mostly being targeted by illegal hunters for bushmeat sale. For example, illegal hunters in the Serengeti prefer to hunt larger animals for their potential higher income returns [20] and such hunting pressure caused the buffalo population to collapse in the 1980s [49]. By contrast, smaller mammals (with higher reproductive

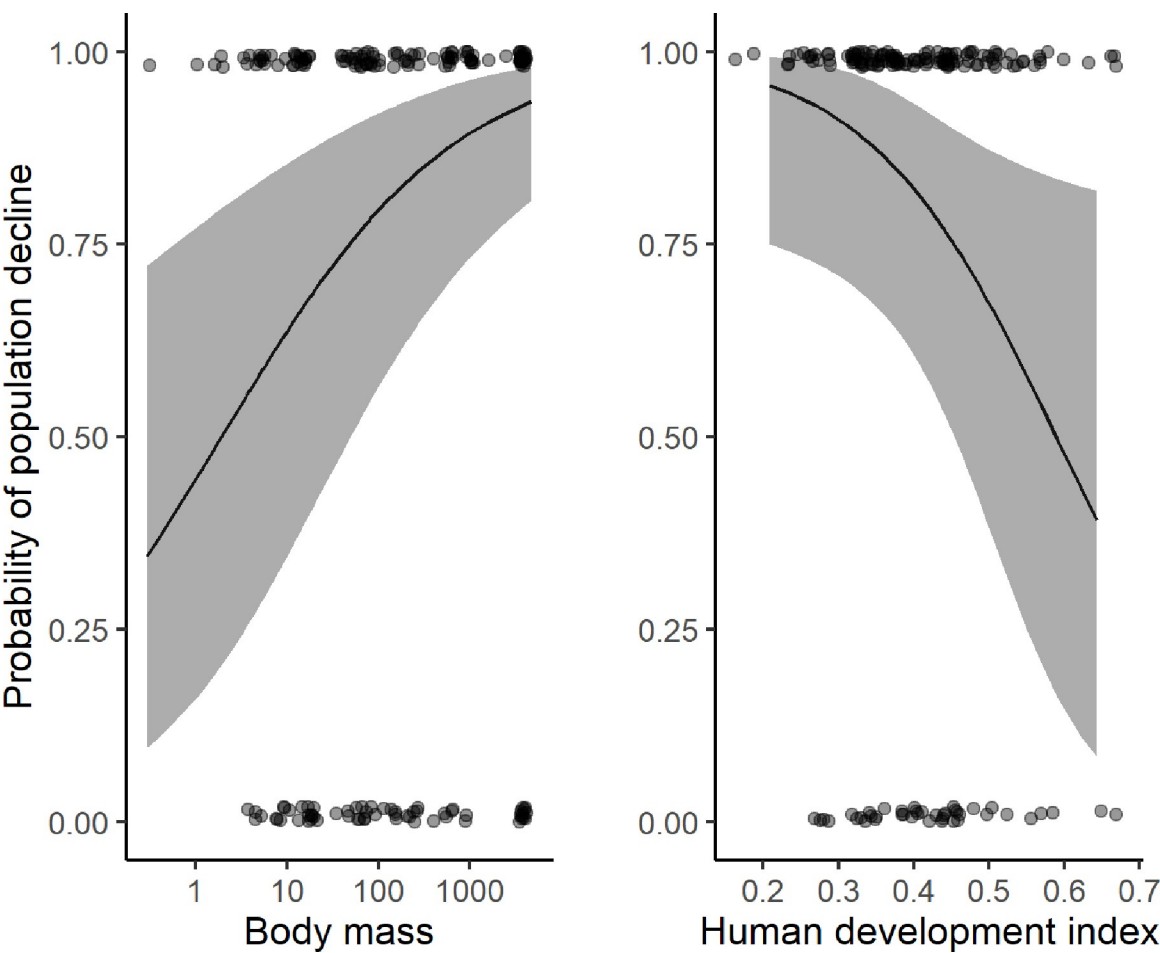

**Fig 4. The probability of decline of mammals in Africa's PAs threatened by illegal hunting pressure showing population decline was strong in PAs located in countries with low human development index and in species with larger body mass.**

and growth rates) showed fewer declines and appeared to sustain harvest, though relatively few small species are the specific targets of illegal hunting in PAs due to their relatively low income returns from bushmeat sale [20]. The pattern of species declines across the network of PAs is worrying and suggests that PA policing (including access to appropriate conservation information) and resources need to be improved. PA-specific information is important for understanding how illegal hunting varies spatially and across time and there is a need to be able to predict future trends and thereby possible future management strategies [62]. Although at a global scale, mammals were less likely to decline by hunting in strict PAs, the opposite was true for Asian PAs. This could be attributable to three reasons; first, it could be that due to high illegal trade of wildlife body parts for traditional medicines in most parts of Asia [63], illegal hunters are forced to enter into protected areas where most sought after species such as snow leopard, tiger, pangolin, orangutans and sub bears still remain in order to satisfy demand for the traditional Asian medicine [63–65]. Second, wild mammals targeted for body parts may have been hunted to completion in wider unconserved landscapes across Asia where human-induced land pressures have increased [57, 66]. Thus, illegal hunters may resort to hunting in protected areas as they remain the only sources of these animals. Third, habitat loss and illegal hunting could be occurring together inside these PAs, hence hastening extinction

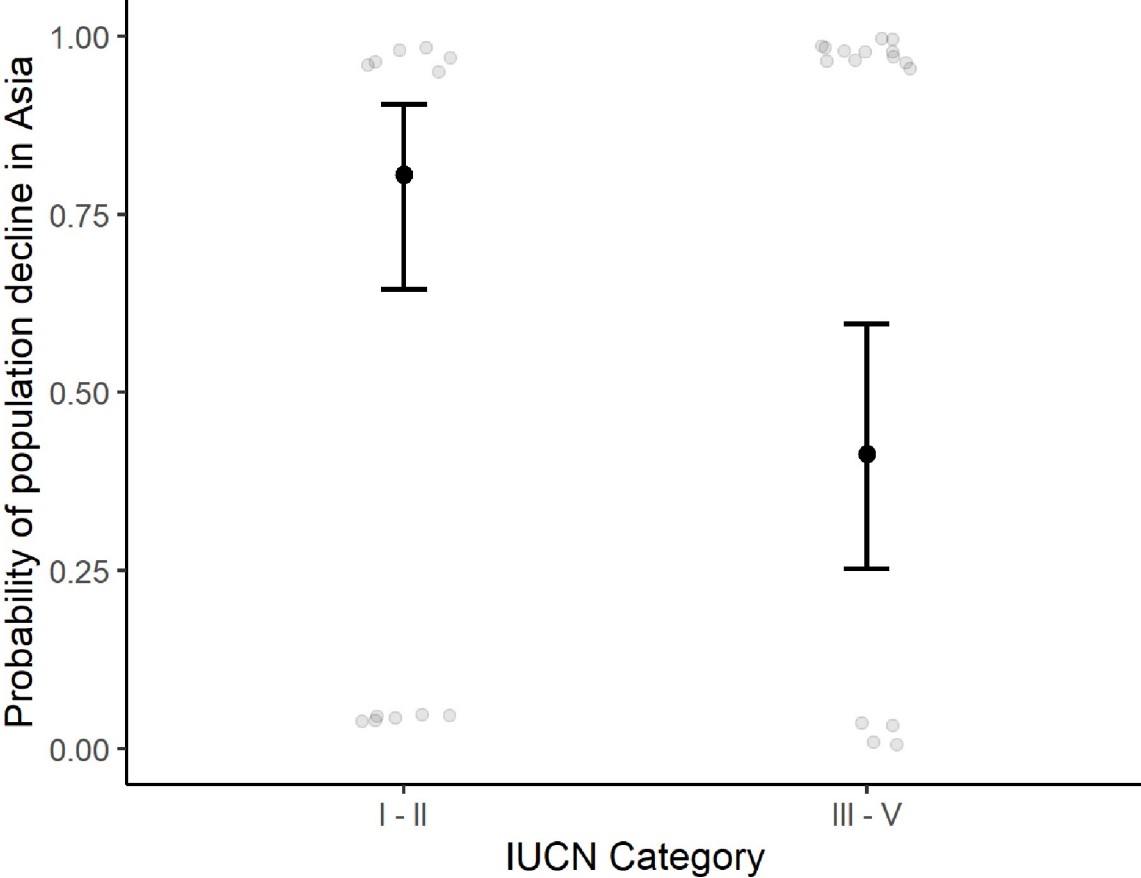

**Fig 5. Probability of population decline for Asia's only PAs indicating strong influence of PA types.** Higher decline of mammals was more likely in stricter than less-strict PAs.

risks for these animals [67, 68]. These results are consistent with previous studies that have reported biodiversity decline and loss within PAs in these regions e.g. Geldmann and others [10], Harrison [69], Laurance and others [70] and Gallego-Zamorano and others [14]. Our findings suggest that the effectiveness of Asian PAs requires urgent attention to improve biodiversity conservation across in continental Asia.

The geographical bias in the spatial distribution of research observed in these data is likely a consequence of interests among the researchers rather than being driven solely by the levels of illegal hunting in particular PAs or countries. However, the temporal and spatial patterns of research observed in this study provide insight into the extent of the problem of illegal hunting in PAs and therefore suggest that PAs are currently in need of new strategies to minimize impacts of illegal hunting pressure and to improve their conservation effectiveness [71]. To date, research effort has concentrated on quantifying the extent and impact of illegal hunting on focal species; in other words, documenting the problem. Far less information is available on which conservation management strategies (including human development and preventing illegal international trade, as well as within-PA activities) are most effective at reducing illegal hunting pressures. New research should focus on developing and testing new methods for reducing levels and impacts of illegal hunting on wild mammal species in PAs.

## Conclusion

Tackling illegal hunting within PAs remains a high conservation priority. Our results suggested that a combination of strategies may be required to reduce the extent of illegal hunting activities. We found that illegal hunting in poor countries often leads to population declines within PAs, suggesting that poverty alleviation may be an appropriate conservation strategy to reduce illegal hunting pressures [44]. The implication of this for local and national policies is that more effort needs to be invested to improve the social and economic status of the human populations. This needs to work in tandem with increasing the effectiveness of traditional conservation activities to prevent illegal hunting; which may itself reduce the inclination of people to attempt future illegal hunting activities. Curbing external pressures to the PAs also needs improved efforts to prevent encroachment and other illegal activities such as crop farming within and squeezing of the PA borders. This will be urgent especially for Asia where stricter PAs are at greatest risk of losing their mammal populations.

## Supporting information

**S1 File.**
(DOCX)

**S1 Raw data.**
(CSV)

## Acknowledgments

We are grateful to the anonymous reviewers for the comments on earlier drafts of this manuscript.

## Author Contributions

**Conceptualization:** Alfan A. Rija.

**Data curation:** Alfan A. Rija.

**Formal analysis:** Alfan A. Rija, Rob Critchlow, Chris D. Thomas, Colin M. Beale.

**Investigation:** Alfan A. Rija.

**Methodology:** Alfan A. Rija.

**Supervision:** Chris D. Thomas, Colin M. Beale.

**Validation:** Rob Critchlow.

**Writing – original draft:** Alfan A. Rija.

**Writing – review & editing:** Alfan A. Rija, Chris D. Thomas, Colin M. Beale.

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
