## [Decision Letter · Decision Letter 0]

7 Feb 2020

PONE-D-19-34271

Spatial distribution and research trend of illegal activities and the factors associated with wild mammal population declines in protected areas

PLOS ONE

Dear DR Rija,

Thank you for submitting your manuscript to PLOS ONE. After careful consideration, we feel that it has merit but does not fully meet PLOS ONE’s publication criteria as it currently stands. Therefore, we invite you to submit a revised version of the manuscript that addresses the points raised during the review process.

We would appreciate receiving your revised manuscript by Mar 23 2020 11:59PM. To enhance the reproducibility of your results, we recommend that if applicable you deposit your laboratory protocols in protocols.io, where a protocol can be assigned its own identifier (DOI) such that it can be cited independently in the future. For instructions see: http://journals.plos.org/plosone/s/submission-guidelines#loc-laboratory-protocols

We look forward to receiving your revised manuscript.

Kind regards,

Stefano Grignolio, Ph.D

Academic Editor

PLOS ONE

Additional Editor Comments (if provided):

Both the reviewers have appreciated your idea and found it intriguing and worth publishing the potential results of this research. There are several aspects, however, that must be carefully addressed before I can consider your paper for publication.

In general, the current draft of the manuscript is confusing, and it is not presented in an attractive way so. The manuscript has to be fully reviewed by a native English speaker. The vocabulary needs a revision (see reviewer #2) to be consistent with the literature and through the text. Both reviewers point out the need to complete the review of literature, including the scientific papers published in the last years (from 2016 to date).

I agree with the concerns of the reviewer #2 on the statistical analysis. Please, pay attention to provide a full reply to the comments about this issue. Moreover, I did not understand your description of the results about figures 6 and 7: I think that these trends are not significant, because the CIs are very large and they include the variations described by the regression lines.

Journal Requirements:

Reviewers' comments:

Reviewer's Responses to Questions

**Comments to the Author**

1. Is the manuscript technically sound, and do the data support the conclusions?

Reviewer #1: Yes

Reviewer #2: Partly

2. Has the statistical analysis been performed appropriately and rigorously? 

Reviewer #1: Yes

Reviewer #2: Yes

3. Have the authors made all data underlying the findings in their manuscript fully available?

Reviewer #1: Yes

Reviewer #2: No

4. Is the manuscript presented in an intelligible fashion and written in standard English?

Reviewer #1: No

Reviewer #2: No

5. Review Comments to the Author

Reviewer #1: Abstract

Line 34 - It should be 1980-2014?

Introduction

Line 75 - change "hunting" to "hunting/poaching"

Line 76 - Insert "threat" in the "each threat can reduce..."

Line 95 - conjugate verbs in the past tense. E.g. change "review" to "reviewed".

Line 96 - 1980-2014?

Line 96 - Why only until 2014? A literature review paper should be updated. It has been 6 years and certainly many studies have been published across the globe and should be included in the analysis before the MS is considered for publication.

Line 98 - change "evaluate" to "evaluated".

Line 99 - We assessed

Line 106 - "Material and methods"

Line 107 - Data collection?

Overall, I recommend that MS be reviewed by an English native prior to publication.

Line 109 - What is the reason for being from 1950? Authors must define why this temporal cut. Also, in the abstract and introduction is written from 1980-2014?

Lines 112-113 - Again, a review study of the scientific literature must be current. Therefore, it is important that the authors include data by 2019 at least.

Lines 147-150 - The MS title is said to be a study of mammalian populations. It may be better to modify for vertebrates since the authors have used other taxonomic groups.

Lines 165-167 - Again, if other taxonomic groups were analyzed, it is important to modify the title of the study.

Line 173 - "referred"

Lines 227-228 - Importantly, the period included in the search was 1950-2014 and not 1980-2014. The fact that no studies were found until 1979 is a result.

Lines 288-289 - ???

Line 293 - ???

Line 333 - ???

Line 339 - ???

Line 360- ???

Line 364 - ???

Line 430 - ???

The MS presents important information to understand how impacts on protected areas are distributed across the globe. And no doubt it is of great relevance to conservation. However, it is necessary to update the data before publication! A literature review study should be as current as possible at the time of publication. That is, authors must at least include data from papers by 2019.

Reviewer #2: The researchers of this paper made a review of the illegal activities affecting populations of different species inside protected areas (PAs). This is a very relevant topic because it is increasingly shown that protected areas are not enough to stop the biodiversity crisis we observe worldwide. Especially, in the last years, it has been shown that overexploitation and land-use change are the main drivers of biodiversity loss. However, the paper does not include any reference to recent studies (the most recent reference is from 2016) and therefore I think they missed some relevant and new insights on the topic. I believe that the manuscript and their database has potential for publication, yet, the manuscript was confusing in many parts, difficult to follow, and not presented in an attractive way so, in my opinion, it requires a profound revision from the authors to clarify several aspects and improve the scientific basis of it.

In general, I think that more attention needs to be paid to the vocabulary and the overall aim of the paper. In the introduction (and title) they talk about illegal activities, however, in the methods they explain that they only included illegal extraction and no other illegal activity as such for the analysis (criteria ii). I would frame the paper differently because otherwise, the reader expects a different outcome from it. Moreover, the methods were really short and some decisions that they made were difficult to understand, e.g., why put all non-mammals together? The dataset although relatively large is too patchily distributed and it should be refined (filtered) before being analysed. I suggest to restrict the analysis to Africa as most of the data (~80%) is from there. In contrast to the methods, the results were quite extensive with some non-informative figures that could be merged or eliminated, tables that could be clearer, and paragraphs that also need clarification (see comments). Overall, the discussion was easier to follow than other parts of the paper, however, as I had difficulties following the general setup and I think that some of the results cannot be fully trusted due to low data points, most of my specific comments regard the methods.

Title

The main title is misleading as it talks about illegal activities, while the paper is about poaching only. Therefore, the short title is much better as it focusses on the poaching impact. Please, consider changing the title

Introduction

Lines 66-71: Please revise recent literature on large-scale patterns on the efficiency of PAs (e.g. Geldmann et al. 2015, 2019) and wilderness areas (Di Marco et al. 2019) to protect biodiversity. Also check the effect of hunting and land use on biodiversity worldwide (Brodie et al. 2015; Ripple et al. 2016; Benítez-López et al. 2017, 2019; Gallego‐Zamorano et al. 2020) because in your citations, you only focus on local assessments but there has been an effort on the last years to assess overall impacts of these pressures. This also applies to other parts of the text e.g. Lines 91-93 and discussion.

Line 96: Both here and in the abstract the years (I supposed) are wrong, please correct.

Methods

Lines 115: Until now, you talked only about illegal activities in general and mentioned illegal extraction (poaching) as one of them. Here, you restricted your search to only studies that included illegal extraction (criteria i and ii), which is fine but mislead the overall aim of the paper.

Line 127-129: What if several studies satisfied all the criteria, how did you select the paper?

Line 134: But logging was not included in the analysis right?

Line 139-140: I would like to know all the information that you put in the database and see the full database that you used for the analysis. Please, remove the “etc” and name all the variables, do not be short in words.

Lines 140-143: Do you use meta-analytical tools for the analysis or not? If yes, which type of weighting method did you use for each study (sample size, variance)? How can I believe that your analyses are robust? Reference 18 is not suitable for this statement.

Line 149: The fish data is missing in SM1, is all the body mass coming from primary sources?

Line 155-162: As you focus on population changes of species inside PAs, could you use spatial analysis to calculate for example the agricultural land-use change inside each of the PAs? I propose this because PAs, in general, are not big and species suffer more from local-scale changes (e.g. deforestation or hunting in the area) than from country-level changes. Some areas of a country can be heavily degraded while others are intact, therefore a more detailed and spatially-explicit analysis will benefit the paper in my opinion.

Lines 164-165: If the purpose of your study is to assess the effectiveness of PAs against illegal activities, specifically illegal extraction, please consider to re-do the levels of PAs and stick to PAs that are designated at conserving biodiversity. Categories I‐IV, are designated for that, with I-II being stricter. Categories V‐VI allow sustainable use (see Santini et al. 2016; Oldekop et al. 2016).

Lines 167-170: This can mislead the results because reptiles, birds, fishes and mollusc have very different ecological requirements and their extraction is very different. If the data was poor, then I would simply not use it. Same applies to the spatial data, I would not include the European study nor (maybe) the American ones.

Lines 171-209: In this paragraph, I would specify more explicitly that the response variable of your models is species decline (1) or not (0). This is mentioned in the previous section but as an example, so here I was a bit confused about which response variable you used.

Line 183: Please be consistent with the vocabulary. Here, you use poaching type, but in line 194 you use hunting level (same applies to the whole paper). Poaching refers to illegal hunting, but the level of hunting is something different, e.g., intense vs not intense hunting. I would not use poaching and hunting interchangeably as hunting can be legal.

Line 188: Which datasets? Do you have more than one?

Lines 195-197: How did you do this? Please, expand the explanation.

Line 202-204: How about using cross-validation instead of a subset data specifically for elephants? What is sufficient data in this case?

Line 204-206: So if I understood correctly you have three sets of data, in all sets, you started from a global model and did backwards removal to assess the generality of the effects, is this correct? Did you use any model selection criteria (AIC or BIC) to assess if the model with all the significant variables was the best or not? Moreover, as I said, I would stick only to the continents with enough data (Africa and Asia) the rest are quite scarce (maybe even only Africa as it accounts for 80% of your data). Figure 1 shows how this is a problem with very few points in Europe and America.

Results

In my opinion, there are too many figures (8 in total) and some of them are not very informative, please, consider to remove some of them or combine them. For instance, Figure 2 could be replaced with text.

Lines 246-252: These two paragraphs could be merged in one, Figure 3 already shows the increase in studies so no need for Figure S1. Moreover, is there a reason for the different colors in Figure 3?

Line 263-264: These numbers also show a strong biased towards mammals with very low numbers for the rest of the groups, therefore, I would suggest to completely remove them from the analysis and not include them as “non-mammals”.

Line 286-292: This paragraph and Table 1 are really confusing. You mix all different models and is really difficult to follow. Why Figure 6b is the model without African elephants? And did you only removed African elephants or all elephants as mentioned in line 200? Again, please be consistent with terminology. Moreover, I don’t understand Model 1, how is it possible that none of the poaching types (not levels) is significant when including the elephants? Aren’t elephants very influenced by poaching (line 202)?

Line 302: I thought Figure 6a is for Model 1 (line 288-289) but in the caption, you put that is for the model without elephants, can you clarify it?

Line 315: I think you confuse the terminology here is not best explained but significantly affected by agricultural change. Could you provide the explained variance by each of the models, i.e. marginal and condition R2?

Line 356: Did you actually removed Europe? How about Central America? Why did you combine Latin America+Asia?

Line 360: So the stricter the more decline?

Discussion

Please, do not talk about illegal activities in general through all the discussion as you only included illegal extraction (poaching) in the analysis, focus on that.

Lines 376-377: Could you maybe create different body size groups to test this?

Line 378: You did not calculate habitat loss as such, even less at the PA scale. Is there any animal in your list that have affinity for agricultural areas? If yes, then the increase in agriculture for them is not habitat loss.

Suggested references

Benítez-López A, Alkemade R, Schipper AM, Ingram DJ, Verweij PA, Eikelboom JAJ, Huijbregts MAJ. 2017. The impact of hunting on tropical mammal and bird populations. Science 356:180–183. Available from: http://science.sciencemag.org/content/sci/356/6334/180.full.pdf.

Benítez-López A, Santini L, Schipper AM, Busana M, Huijbregts MAJ. 2019. Intact but empty forests? Patterns of hunting-induced mammal defaunation in the tropics. PLOS Biology 17:e3000247. Available from: https://doi.org/10.1371/journal.pbio.3000247.

Brodie JF, Giordano AJ, Zipkin EF, Bernard H, Mohd-Azlan J, Ambu L. 2015. Correlation and persistence of hunting and logging impacts on tropical rainforest mammals. Conservation Biology 29:110–121. John Wiley & Sons, Ltd. Available from: https://doi.org/10.1111/cobi.12389.

Di Marco M, Ferrier S, Harwood TD, Hoskins AJ, Watson JEM. 2019. Wilderness areas halve the extinction risk of terrestrial biodiversity. Nature 573:582–585. Available from: https://doi.org/10.1038/s41586-019-1567-7.

Gallego‐Zamorano J, Benítez‐López A, Santini L, Hilbers JP, Huijbregts MAJ, Schipper AM. 2020. Combined effects of land use and hunting on distributions of tropical mammals. Conservation Biology:cobi.13459. Available from: https://onlinelibrary.wiley.com/doi/abs/10.1111/cobi.13459.

Geldmann J et al. 2015. Changes in protected area management effectiveness over time: A global analysis. Biological Conservation 191:692–699. Available from: http://www.sciencedirect.com/science/article/pii/S0006320715300793.

Geldmann J, Manica A, Burgess ND, Coad L, Balmford A. 2019. A global-level assessment of the effectiveness of protected areas at resisting anthropogenic pressures. Proceedings of the National Academy of Sciences:201908221. Available from: http://www.pnas.org/content/early/2019/10/22/1908221116.abstract.

Oldekop JA, Holmes G, Harris WE, Evans KL. 2016. A global assessment of the social and conservation outcomes of protected areas. Conservation Biology 30:133–141. John Wiley & Sons, Ltd. Available from: https://doi.org/10.1111/cobi.12568.

Ripple WJ et al. 2016. Bushmeat hunting and extinction risk to the world’s mammals. Royal Society open science 3:160498. The Royal Society. Available from: https://www.ncbi.nlm.nih.gov/pubmed/27853564.

Santini L, Saura S, Rondinini C. 2016. Connectivity of the global network of protected areas. Diversity and Distributions 22:199–211. Available from: http://doi.wiley.com/10.1111/ddi.12390.

6. PLOS authors have the option to publish the peer review history of their article (what does this mean?). If published, this will include your full peer review and any attached files.

Reviewer #1: No

Reviewer #2: No

---

## [Author Response · Author response to Decision Letter 0]

16 Apr 2020

Details of our responses to each reviewer and the Journal Editor is found in the Response to Reviewers document included in the attachments

---

## [Decision Letter · Decision Letter 1]

26 May 2020

PONE-D-19-34271R1

Global extent and drivers of mammal population declines in protected areas under illegal hunting pressure

PLOS ONE

Dear Dr. Rija,

Thank you for submitting your manuscript to PLOS ONE. After careful consideration, we feel that it has merit but does not fully meet PLOS ONE’s publication criteria as it currently stands. Therefore, we invite you to submit a revised version of the manuscript that addresses the points raised during the review process.

We look forward to receiving your revised manuscript.

Kind regards,

Stefano Grignolio, Ph.D

Academic Editor

PLOS ONE

Additional Editor Comments (if provided):

I appreciated the impressive job of responding to comments and increasing the quality of the manuscript. Both reviewers agree to point out this result, even if the reviewer #1 suggests to add some articles in your review.

Although I noted the big efforts to improve the writing of the manuscript, I agree with reviewer #1 to request you a further revision in the use of specific terms and acronyms.

I feel that the method section needs a more detailed description of the statistical analysis, particularly of the methods used to estimate the explanatory variables. Finally, I would like to see a broader description of the results in order to better highlight the findings of the manuscript and to make them easier to understand for readers.

Reviewers' comments:

Reviewer's Responses to Questions

**Comments to the Author**

1. If the authors have adequately addressed your comments raised in a previous round of review and you feel that this manuscript is now acceptable for publication, you may indicate that here to bypass the “Comments to the Author” section, enter your conflict of interest statement in the “Confidential to Editor” section, and submit your "Accept" recommendation.

Reviewer #1: (No Response)

Reviewer #2: All comments have been addressed

2. Is the manuscript technically sound, and do the data support the conclusions?

Reviewer #1: Partly

Reviewer #2: Yes

3. Has the statistical analysis been performed appropriately and rigorously? 

Reviewer #1: Yes

Reviewer #2: Yes

4. Have the authors made all data underlying the findings in their manuscript fully available?

Reviewer #1: Yes

Reviewer #2: Yes

5. Is the manuscript presented in an intelligible fashion and written in standard English?

Reviewer #1: No

Reviewer #2: Yes

6. Review Comments to the Author

Reviewer #1: The second version of the MS submitted by the authors presented a text that was difficult to understand, which complicated decision making. The study is important, however, it needs a careful editing of the text to improve reading. There is a lack of standardization in writing and abbreviations, terms, etc. Furthermore, I believe that the review conducted by the authors is incomplete and biased, because several studies conducted in Brazil that showed the impact of illegal hunting on mammals have not been mentioned.

My biggest concern is the writing of the MS and the absence of some important references.

See my comments in the PDF file.

Reviewer #2: (No Response)

7. PLOS authors have the option to publish the peer review history of their article (what does this mean?). If published, this will include your full peer review and any attached files.

Reviewer #1: No

Reviewer #2: No

---

## [Author Response · Author response to Decision Letter 1]

17 Jul 2020

Many thanks to the reviewers for their constructive comments on the manuscript. We have responded to each comment from the reviewers and Editor. The details can be found in the response to reviewers document attached

---

## [Decision Letter · Decision Letter 2]

30 Jul 2020

Global extent and drivers of mammal population declines in protected areas under illegal hunting pressure

PONE-D-19-34271R2

Dear Dr. Rija,

We’re pleased to inform you that your manuscript has been judged scientifically suitable for publication and will be formally accepted for publication once it meets all outstanding technical requirements.

Kind regards,

Stefano Grignolio, Ph.D

Academic Editor

PLOS ONE

Reviewers' comments:

Reviewer's Responses to Questions

**Comments to the Author**

1. If the authors have adequately addressed your comments raised in a previous round of review and you feel that this manuscript is now acceptable for publication, you may indicate that here to bypass the “Comments to the Author” section, enter your conflict of interest statement in the “Confidential to Editor” section, and submit your "Accept" recommendation.

Reviewer #1: All comments have been addressed

2. Is the manuscript technically sound, and do the data support the conclusions?

Reviewer #1: Yes

3. Has the statistical analysis been performed appropriately and rigorously? 

Reviewer #1: Yes

4. Have the authors made all data underlying the findings in their manuscript fully available?

Reviewer #1: Yes

5. Is the manuscript presented in an intelligible fashion and written in standard English?

Reviewer #1: Yes

6. Review Comments to the Author

Reviewer #1: After careful review of the second version of the MS entitled PONE-D-19-34271R2 "Global extent and drivers of mammal population declines in protected areas under illegal hunting pressure", I decided to recommend for publication.

7. PLOS authors have the option to publish the peer review history of their article (what does this mean?). If published, this will include your full peer review and any attached files.

Reviewer #1: No

---

## [Editor Report · Acceptance letter]

11 Aug 2020

PONE-D-19-34271R2 

Global extent and drivers of mammal population declines in protected areas under illegal hunting pressure 

Dear Dr. Rija:

I'm pleased to inform you that your manuscript has been deemed suitable for publication in PLOS ONE. Congratulations! Your manuscript is now with our production department. 

Kind regards, 

on behalf of

Dr. Stefano Grignolio 

Academic Editor

PLOS ONE